# Exploiting Spatial-Temporal Data in Knowledge Graphs for Enhanced Prediction

## ABSTRACT

Knowledge graphs (KGs) have been increasingly employed for link prediction and recommendation using real-world datasets. However, the majority of current methods rely on static data, neglecting the dynamic nature and the hidden spatial-temporal attributes of real-world scenarios. This often results in suboptimal predictions and recommendations. Although there are effective spatial-temporal inference methods, they face challenges such as scalability with large datasets and inadequate semantic understanding, which impede their performance. To address these limitations, this paper introduces a novel framework for constructing and exploring spatial-temporal KGs. Our approach seamlessly integrates spatial and temporal data to form KGs. These KGs are further exploited through a new 3-step embedding method. Output embeddings can be used for future temporal sequence prediction and spatial information recommendation, providing valuable insights for various applications such as retail sales forecasting and traffic volume prediction. By integrating spatial-temporal data into KGs, our framework offers a more comprehensive understanding of the underlying patterns and trends, thereby enhancing the accuracy of predictions and the relevance of recommendations. This work paves the way for more effective utilization of spatial-temporal data in KGs, with potential impacts across a wide range of sectors.

## KEYWORDS

Knowledge graph, spatial-temporal data

**ACM Reference Format:**
Anonymous Author(s). 2023. Exploiting Spatial-Temporal Data in Knowledge Graphs for Enhanced Prediction. In *Proceedings of The 2024 ACM Web Conference (WWW '24).* ACM, New York, NY, USA, 9 pages. https://doi.org/XXXXXXX.XXXXXXX

## 1 INTRODUCTION

Knowledge graphs (KGs) are directed graphs comprising entities (nodes), their attributes, and the relationships between them. They represent information as facts using a node-edge-node structure. For instance, the triplet (Macdonald-compete-Burger King) represents a competitive relationship between Macdonald and Burger King. KGs adeptly capture intricate relationships between entities, enabling more contextually rich and accurate predictions. By encoding millions of real-world events or facts into graphs, KGs

facilitate various downstream tasks such as recommendation system [23], information retrieval [9], and question answering [14]. Although sometimes KG construction suffers from incompleteness due to insufficient construction criteria or data scarcity, *knowledge graph completion* (KGC) methods fill this gap. These methods infer missing facts based on existing ones in KGs. They learn the embedding of entities and relations on known facts and apply score functions on all possible facts to compute the possibility the fact exists, KG embedding models, like transE [2], to help enhance the comprehensiveness and utility of the KG.

Relying solely on static knowledge graphs for every real-world dataset proves insufficient. In practical scenarios, such as retail sales and traffic, historical facts influence potential future relations. Spatial-temporal data, inherently dynamic and complex, exhibits dependencies and relationships that evolve across time and space. The dynamic features of the data complicate the construction and maintenance of KGs that represent data comprehensively and factor in geographical relationships between entities. Static KGC methods treat facts as time-independent, leading to relation and entity embeddings stagnant, which is unrealistic [5]. Many methods are raised towards temporal KG construction and completion [18, 1, 12, 19], but training them on benchmark datasets like Wikidata [17, 16] or YAGO15K [10] proves time-consuming [3]. The time cost is magnified when applied to extensive real-world datasets.

Without using KGs, a myriad of spatial-temporal prediction and recommendation methods have been proposed, yielding promising outcomes across various tasks [25]. Traditional approaches, the statistical and machine learning methods like ARIMA [28], have been complemented by more recent deep learning methods, notably graph convolutional networks [32], have emerged. Despite their effectiveness in specific scenarios, these methods still harbor notable limitations. For instance, they often struggle when capturing the intricate, non-linear relationships endemic to spatial-temporal data, and may fall short of incorporating broader contextual information. Data sparsity posed another challenge, constraining the improvement of their recommendation performance [4]. In contrast, KGs can alleviate this issue, courtesy of their rich semantics information.

Given the outlined challenges, building and exploiting Spatial Temporal Knowledge Graphs (STKGs) presents a promising avenue. For clarity, Fig 1 showcases an STKG tailored for the physical store sales within a city. While various retail outlets like Walmarts, IKEA, are depicted alongside entities from different sectors, such as McDonald. Their sales records, inherently temporal, are typically tabulated over a time interval $\Delta T$, including records in several time intervals $\Delta t_{1,2,...}$. In a static KG, entities might be linked through relations like *competing*, *collaborating*, etc. However, in a temporal KG, these relations might evolve over time based on sales data or other practical considerations. Beyond temporal aspects, entities also exhibit geographical relations, heavily influenced by *locations* and *distances* separating them.

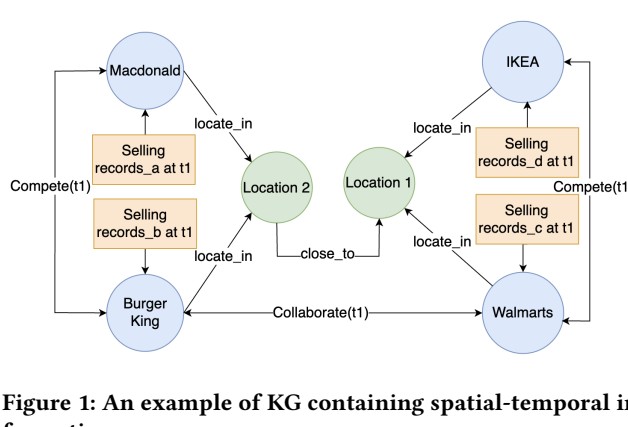

**Figure 1: An example of KG containing spatial-temporal information**

Entities within STKGs can be expressed using triplets, $(Shop, t_i, loc)$, signifying an entity's state at a specific time. Meanwhile, relations are represented as $(e_i, relation, e_j)$ under a given $time, location(i, j)$ highlighting the spatial-temporal connection between two entities. By integrating these triplets, a semantic path is constructed, elucidating the evolution of relationships grounded in spatial-temporal data.

STKGs are versatile tools suitable for predictive tasks, encompassing forecasting specific store sales or traffic volumes at designated stations. Their utility also extends to recommendation tasks, facilitating predicting the next POI. Despite the potential, the dynamic nature of data and intricate relationships within the graph present challenges in harnessing an STKG effectively for downstream applications, which leads to research questions: How can an STKG framework be versatile enough to accommodate diverse data types? How to enable this framework for KG completions while ensuring its interpretability?

In this paper, a novel framework is raised for constructing and exploring Spatial-temporal knowledge graphs for prediction and recommendation. By integrating spatial-temporal data into KGs and exploiting these KGs through entity and relation embeddings, the framework aims to leverage the strengths of KGs to enhance the accuracy and relevance of spatial-temporal predictions. While keeping the embeddings suitable for real-world datasets, the framework ensures efficiency as well as interpretability. To validate its efficacy, the framework was applied to two datasets: Safegraph's Spend-Ohio dataset and the traffic volume of New South Wales (TFNSW) dataset to do experiments on temporal sequence prediction.

## 2 RELATED WORK

### 2.1 Spatial-Temporal Data Prediction & Recommendation

Doing prediction or recommendation on spatial-temporal data has been a hot topic of research in recent years, addressing challenges in domains like traffic volume prediction, selling record prediction, and next point-of-interest recommendation. Early methods are based on statistical knowledge, or using machine learning. ARIMA [28] was based on the Wold decomposition theorem, using linear time-series techniques, however, it is unable to capture enough spatial-temporal dependencies, this drawback also occurs in early

machine learning methods, like CSVM [21] and SVM [13]. More recently, researchers have begun to consider more complex methods, thanks to the emergence of deep learning. Recurrent Neural Networks(RNNs) like long short-term memory (LSTM) networks, or Temporal Convolutional Networks(TCNs) can effectively capture temporal features, while Graph Neural Networks(GNNs) and Convolutional Neural Networks(CNNs) perform well in spatial data. The fusion of deep learning models leads to a great fit for spatial-temporal data. Spatial-temporal GNNs like graph convolutional neural networks(GCNs) can simultaneously model spatial and temporal information [7], and are widely used in real-world cases like traffic flow prediction [8, 29, 27] or weather forecasting [15].

### 2.2 Knowledge Graph for prediction

*2.2.1 Static KGs for Prediction.* Since KGs have a unified structure, based on their embeddings or paths, they can be used to predict potential links hidden in established datasets. For static data, KGs can assist and accelerate drug discovery [31] in the medical field, and they also perform well on fake news detection [6] by finding the shortest path between facts.

*2.2.2 Temporal KGs/STKGs for Prediction.* Dynamic data, typically sourced from sensors, can also be transferred into structured entities and shaped into temporal KGs(TKGs) or STKGs. Embeddings encompassing distinct spatial or temporal information are compared to determine the entities that would appear in certain time points under certain locations. Since dynamic KGs capture time relationships between entities in events, temporal predictions like the time of natural disasters [11] could be achieved. STKGs also help in spatial predictions, by modeling trajectories data, users' mobility patterns or activities can be predicted [24, 4].

### 2.3 Knowledge Graph for recommendation

KGs can help solve cold start problems as external sources. Also, by modeling data into KGs, Problems caused by the sparsity or low popularity bias of data are reduced, which can normally influence traditional recommendation methods like collaborative filtering. Normally the recommendation methods on KGs are categorized into path-based and embedding-based.

*2.3.1 Path-based recommendation.* Paths in KGs contain relationships between entities, enabling the extraction of features such as users' preferences or item characteristics by analyzing paths. KPRN [26] used the LSTM network to represent path information, like users and movie interactions, thus can calculate user preferences towards target movies.

*2.3.2 Embedding-based recommendation.* Entities and relations can normally be transferred into embeddings under certain rules, these embeddings can be applied to recommendation algorithms. Entity2rec [20] uses property-specific embeddings on KGs to do recommendation, while HAKG [22] uses subgraph embeddings for enhanced user preference prediction.

While the aforementioned methodologies have registered good performances in designated tasks, they are encumbered by certain limitations: 1) Their inherent complexity or the extensive versatility of entity types often renders them time-intensive or restricts their adaptability to diverse domains. 2) They are not explainable enough

to describe features extracted from spatial-temporal data. The proposed model aspires to bridge these gaps, presenting a solution that is streamlined and adaptable to diverse data types while ensuring interpretability.

## 3 METHODOLOGY

### 3.1 Preliminaries and method overview

The STKG problem is defined as: An optimal STKG should accommodate the dynamic nature of data, adapting to changes in entities' attributes influenced by time and location. Moreover, it is essential for the framework to facilitate STKG completions post-construction and predict forthcoming attributes.

**Input representation** The objective of the proposed STKG is to attain universality. To this end, a uniform representation for diverse types of spatial-temporal data is integrated to generalize raw entities and relations types.

**STKG embedding model** The embedding model is designed to encode entity attributes into vector representations and subsequently decode embeddings into numerical representations mirroring the raw data. The embedding model facilitates KG completion on existing STKG and enables the prediction of underlying or between entities.

Table 1 summarizes notations used in the paper as well as their meanings.

#### Table 1: Notations and descriptions

| Notation | Description |
| --- | --- |
| e, r | An entity and a relation |
| **e**, **r** | Vector representation of e and r |
| $r_{i,j}$ | directional relation from i to j |
| E, R | Entity set and relation set |
| $d(e_i, e_j)$ | Distance between two entities |
| G | A STKG |
| T | The set of time |
| $e_{attribute}$ | The embeddings of certain attribute of entities |

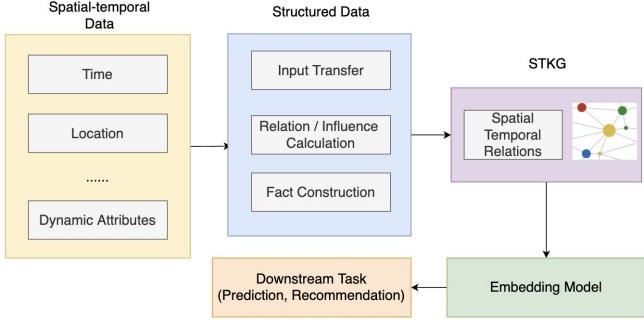

**Figure 2: The workflow of proposed framework**

Figure 2 illustrates the general workflow of the framework. Upon acquiring spatial-temporal data, pre-established rules are utilized

to extract and compute entities, relationships, and facts, thereby constructing STKG while *ensuring limited entity types and relationship types*. Subsequently, a new embedding model is raised to vectorize the features of entities, enabling the utilization of the STKG in downstream tasks. This streamlined process facilitates a more efficient and effective application of knowledge graphs in real-world scenarios, able to be used for inference with enhanced speed. Finally, the underlying patterns and insights captured by the STKG are interpreted based on its structure, making the whole model explainable.

### 3.2 Knowledge Graph Construction

*3.2.1 Definition of STKG.* The spatial-temporal knowledge graph is defined as graphs $G = (E, R, T, F)$, where $E$, as table 1 shows, is entities that contains spatial-temporal attributes. $R$ represent the set of relation between entities. $T$ describes how the temporal records get divided. $F$ is the set of facts mentioned in section 1. Specifically, $R$ and $T$ in the knowledge graph define *certain relation between entities under certain time*, which denotes facts. Facts under STKG are seen as a quadruple $(e_i, t, t_j, r)$.

*3.2.2 Simplified STKG.* Considering when entities like stores are rigidly classified according to their business establishments, as exemplified by the 6-digit North American Industry Classification System (NAICS) code. The strict categorization can lead to an excessive fragmentation of entity types. Also, the dynamics of relationships between entities can vary significantly based on spatial and temporal factors. Two entities, even if their spatial distances are fixed, might have totally inverted relations at different times. Moreover, detailed numerical time and location are hard to be transferred as distinct entities.

In light of these complexities, the simplified STKG aims to provide a more flexible and realistic representation of entities and their relationships, establishing rules for the SSTKG as follows:

- *Rule 1:* Time and location are not treated as independent entities. Instead, they are integrated as attributes inherent and between entities, represented as part of entity and relation embeddings.
- *Rule 2.* The model prioritizes a reduction in the number of entity types, embedding classification data directly within the entity. This not only simplifies the graph structure but also facilitates more efficient and direct retrievals of classification information from the entity embeddings.
- *Rule 3.* Numerical representations are adopted to directly articulate the relationship between two entities. Under this paradigm, the association between entities is conceptualized as a continuous variable termed "influence". Within this framework, any pair of entities can exhibit a relationship that is fluid across both temporal and spatial dimensions
- *Rule 4:* Relationships between entities that are quantitatively negligible are omitted, ensuring focus on significant interactions and reducing noise within the graph.

Leveraging this SSTKG framework, entities are directly extracted from structured data. The process of relation extraction is thus transformed into "relation computation", or "influence computation", while fact still be seen as the quadruple $(e_i, t, t_j, r)$.

*3.2.3 Algorithm for constructing SSTKG.* The detailed process of constructing the SSTKG is elucidated according to 3.2.2. The temporal records for an entity are viewed as *The result of related entities applying influence plus itself's basic record*, which is:

$$p_{e_0} * Record_{e_0}(t) = \Sigma_{i=1}^n I_{(e_i,e_0)} Record_{e_i}(t) \tag{1}$$

Fitting Equation (1) is seen as a regression process, where 1-p is seen as a parameter quantifying the self-influence of an entity, providing a measure of how much an entity's characteristics contribute to its own behavior or status within the knowledge graph. While temporal variable t represents a time slot, the integration of temporal data and spatial relationships facilitates the computation of a relation "weight":

$$W_{(e_i,e_0)} = \frac{OverallRecord(e_i)}{OverallRecord(e_0)} * log(1 + \frac{\Sigma_{j=i}^n Distance(e_j,e_0)}{n * Distance(e_i,e_0)}) \tag{2}$$

$W_{(e_i,e_0)}$, using overall record and distance between two entities, is seen as a ratio of properties of $e_i$ to $e_0$ Then the p in Equation (1) is counted as:

$$p_{e_0} = \frac{\Sigma_i W_{(e_i,e_0)}}{\Sigma_{k,j} W_{(e_k,e_j)}} \tag{3}$$

Then the influence that entity $e_i$ may apply on $e$ during time slot t is seen as:

$$I_{(e_i,e_0)} = regressionFactor * W_{(e_i,e_0)} \tag{4}$$

Algorithm 1 shows the pseudocode for constructing SSTKG,

---

**Algorithm 1** Constructing a SSTKG using time-series records data

---

**Require:** Entity $E$, Location $L$, time-series records $TS$, distance threshold $D$
**Ensure:** Quadratic relation set $R$
1: **for** $e \in E$ **do**
2:     filtering $E_0 \subseteq E$ where
3:     **for all** $e_i \in E_0$ **do**
4:         **if** Distance$(e, e_i) \leq D$ **then**
5:         **end if**
6:     **end for**
7:     **for** $e_i \in E_0$ **do**
8:         $W_{(e_i,e)} \leftarrow$ Compute weight using ((2))
9:     **end for**
10:    $p_e \leftarrow$ Compute $p$ using ((3))
11:    $influence_{(E_0,e)} \leftarrow$ Compute influence using (1)
12: **end for**

---

In determining the "influence", only the spatial-temporal information of entities is considered. Attributes of entities, such as categories, remain unaddressed. Such an omission in SSTKG construction arises from potential complexities in the data; for instance, the prevalence of numerous categories as seen with the NAICS code shown in Section 3.2. On the other hand, some data is hard to fit entities in specific categories, like traffic volume data. Hence, these data are integrated into KG embedding, as elaborated in Section 3.3.

## 3.3 Embedding Model

One entity's temporal data record as well as its spatial location is assumed to influence other entities' temporal records. While the numerical "influence" is seen as a relation, the embedding model aims to map attributes of entities and relations into low-dimensional vectors. Embeddings generated by the model are further implemented into downstream work. Specifically, the embeddings are categorized into 3 boxes:

*3.3.1 Static Embedding.* This component encapsulates the static attributes of an entity, yielding a representation that remains invariant over time. Static attributes are left when calculating "influence". However, in the computation of the static embedding, these attributes that were previously set aside are reintegrated. Apart from categorical attributes, a summary of the entity's comprehensive spatial-temporal data is integrated into the static embedding. Metrics such as average sales volume or average traffic flow are included to represent the "magnitude" or "scale" of the entity. Equation (5) shows the formation of static embedding, where $\phi$ manages to regularize overall records into a smaller range.

$$e_{i\_static} = e_{i\_category} * \phi(overall\_records) \tag{5}$$

*3.3.2 Dynamic Embedding.* Dynamic embedding contains directions of entity relationships, formed by two subsets: out-embedding and in-embedding

Out embedding signifies the potential influence an entity may impart upon its linked entities. It is configured as the dynamic embedding representing the "influence level" of the entity itself, disregarding spatial relationships with other entities. The computation of the out embedding is shown in Equation (6), encompassing concatenation of the static embedding with its temporal records.

$$e_{i\_out}^t = \psi(e_{i\_static}, e_{i\_records}^t) \tag{6}$$

In Embedding quantifies the influence that an entity receives from its associated entities, reflecting the cumulative impact of these relationships on the entity. Analogously, in the formation of the SSTKG, the embedding is viewed as an aggregate of the entity's inherent influence and the influences exerted by its associated entities. Shown in Equation (8), p is the weight shown in Equation (3).

$$p_i * e_{i\_out}^t = e_{i\_in}^t \tag{7}$$

$$\mathbf{e_{i\_in}^t} = \Sigma_j F(Influence_j * \mathbf{e_{j\_out}^t}) \tag{8}$$

On vector space which is:

$$\mathbf{e_{i\_in}^t} = \Sigma_j(Influence_j * \mathbf{e_{j\_record}^t} + \mathbf{e_{j\_static}}) \tag{9}$$

*3.3.3 Embedding Training Algorithm.* The output of Equation (8) in the embedding model is not directly ascertainable, since after adding the influence, out-embedding needs to be trained to fit the equation, which leads to modification in static embedding. The static embedding and out-embeddings are used as input, optimized embeddings are obtained after training.

Let $E_0$ represent a set of out embeddings of entities that have potential relations, according to SSTKG, with entity $e_0$, while set $R$ denotes the initial influence of entities in $E_0$ as (1*n) vector to $e_0$.

**Table 2: Time cost for training SSTKG on Spend-Ohio dataset**

| entity number | time records (day) | average time(s) |
|---|---|---|
| 1000 | 30 | 347.7 |
| 30000 | 30 | 13087.3 |

Given a training tuple x = $(e_0, R, E_0, t)$, the score function is defined as:

$$f_{p1}(x) = f_1(e_0, R, E_0, t) = ||p_{e_0} * e_{0\_out}^t - e_{0\_in}^t||_2^2 \quad (10)$$

Equation (10) defined a score as how precise one entity is **influenced** by related entities. Meanwhile, valid relations and embedding sets are used to obtain a lower score of $f_{SSTKG}$, then the first loss function is defined as:

$$l_{emb}(x) = l_{emb}(e_0, R, E_0, t) = -\Sigma_i log\sigma(f_{p1}(e_0, R^i, E_0, t) - f_{p1}(x)) \quad (11)$$

Embedding $e_{out}^i$ is replaced in $f_{p2}(e_0, R^i, E_0, t)$ with another random entity that has similar overall selling records in the whole dataset and without relations with $e_i$, using the same Influence value.

An alternate score function is defined for the loss of entity influence values. For an entity $e_0$, its influence on the SSTKG, which is related to entities $E_0$ that connect with $e_0$, $R$ now denotes after optimizing out-embeddings, the influence of $e_0$ to entities in $E_0$. Given a training tuple x = $(e_0, R, E_0, t)$, the score function is articulated as:

$$f_{p2}(x) = f_{p2}(e_0, R, E_0, t) = ||p_{e_0} * e_{0\_out}^t - \Sigma_i R_i * e_{i\_out}^t||_2^2 \quad (12)$$

$$l_{inf}(x) = l_{inf}(e_0, R, E_0, t) = -\Sigma_i log\sigma(f_{p2}(e_0, R, E_0^i, t) - f_{p2}(x)) \quad (13)$$

In $f_{SSTKG}(e_0, R^i, E_0, t)$ one related entity's influence is replaced to average. The second loss function denotes the loss of specific **"influence"** value, which is the relations.

The process of learning improved embeddings and influences are shown as pseudocode in Algorithm 2.

## 4 MODEL PROPERTIES

### 4.1 Effciency and Speed

The proposed model is designed with computational efficiency in mind. It requires less computational resources compared to traditional models, thereby enabling faster construction of the STKG. This feature is particularly beneficial in scenarios where rapid knowledge graph construction is crucial. Here is the test result of constructing and optimizing an SSTKG using the Spend-Ohio dataset mentioned in Section 5.1, with 100 training epochs.

### 4.2 Inference Patterns

By using the embedding model in Section 3.3, a certain entity's temporal record is predicted using its related entities' records. Based on Equation (8), trained static embedding of related entities and their current temporal records are used to compute the target entity's out-embedding. Therefore, final temporal records are decoded from

---

**Algorithm 2** Training entity embeddings and relations for SSTKG

**Require:** $N_{epochInf}$, $N_{epochEmb}$, SSTKG $G$ with initialized $e_{static}$, $e_{out}$, influence
**Ensure:** SSTKG with trained $e_{out}$, influence
1: **for** $i = 1$ to $N_{epochInf}$ **do**
2:     $S_1 \leftarrow G$
3:     **while** $S_1 \neq \emptyset$ **do**
4:         Sample batch $S_{batch} \subset S_1$
5:         $S_1 \leftarrow S_1 \setminus S_{batch}$
6:         $L_1 \leftarrow 0$
7:         **for** $s \in S_{batch}$ **do**
8:             $f_{p1}(s) \leftarrow$ compute score using (10)
9:             $l_{inf}(s) \leftarrow$ compute loss using (11)
10:            $L_1 \leftarrow L_1 + l_{inf}(s)$
11:         **end for**
12:         Update out embeddings using $\nabla L_1$
13:     **end while**
14: **end for**
15: **for** $i = 1$ to $N_{epochEmb}$ **do**
16:     $S_2 \leftarrow G$
17:     **while** $S_2 \neq \emptyset$ **do**
18:         Sample batch $S_{batch} \subset S_2$
19:         $S_2 \leftarrow S_2 \setminus S_{batch}$
20:         $L_2 \leftarrow 0$
21:         **for** $s \in S_{batch}$ **do**
22:             $f_{p2}(s) \leftarrow$ compute score using (12)
23:             $l_{emb}(s) \leftarrow$ compute loss using (13)
24:             $L_2 \leftarrow L_2 + l_{emb}(s)$
25:         **end for**
26:         Update influence in relations using $\nabla L_2$
27:     **end while**
28: **end for**

---

out embedding as well as the static embedding, since for the trained embeddings, influence$\in R$ are obtained, while having related entities' records on time slot $t_1$, the out/in embeddings for $e_0$ is inferred based on Equation (8) and (9). Subsequently, the referred $e_{i\_records}^t$ is decoded in accordance with Equation (6).

### 4.3 Interpretability

Another significant advantage of SSTKG is its interpretability. The simplified structure and the numerical representation of relationships make it easier to understand the underlying patterns and insights captured by the STKG. This interpretability enhances the model's usability, especially in applications where understanding the reasoning behind predictions is important.

Embedding directly reflects the spatial-temporal properties of each entity based on backward induction. The whole fitting and training process, to simply explain, is a process of finding proper embeddings that incorporate an entity's spatial-temporal data, such that the embedding (out-embedding), is viewed as the result of the combined effects of related entities' embeddings(out-embedding), during which the unidirectional relation between two entities serves as the parameter of fitting the whole equation. First, an expansion of the Equation (8) is resented:

$$p_i * e^t_{i\_in} = \Sigma_j \psi(e_{i\_static}, e^t_{i\_records}) * Influence_{j,i} \quad (14)$$

Which could be transferred as

$$p_i * e^t_{i\_in} = \Sigma_j e_{j\_static} * \Omega(e^t_{i\_records}, Influence_{j,i}) \quad (15)$$

Clearly, $\Omega$ after this transformation, served as connecting parameters of out-embeddings (the temporal record) to the influence variables: it's a temporal relation of entity $j$ to $i$, which is further explained as **entity $j$'s influence to $i$ under time** $t$, also it can serve as generating an embedding of temporal relation, which is simplified as:

$$p_i \cdot e^t_i = \sum_j e^t_j \cdot influence_{j,i} = \sum_j e_{j\_static} \cdot r^t_{j,i} \quad (16)$$

Thus, from the final result, training the embedding serves to refine the processes undertaken during SSTKG construction. it optimizes the whole SSTKG, forming the exact relationship using both entities' categorical, spatial, and temporal attributes.

## 5 EXPERIMENTS

### 5.1 Datasets

Two datasets are used to evaluate the performance of SSTKG. The first one is Spend-Ohio data from January 2022 to April 2023, collected by Safegraph, containing many Ohio stores' geographical and categorical information, as well as the selling records **counted by day**. The second one is Traffic Volume of Transport for New South Wales (TFNSW) data, which encompasses the traffic volume from a collection of permanent traffic counters and classifiers in Sydney, with data collated since 2008 on an **hourly basis**. Locations of these counters have been further categorized based on their respective suburbs. Table 3 presents the size of the two datasets. Notably, the 'distance' attribute represents the distance threshold employed during SSTKG construction as per Algorithm 1."

**Table 3: Quantities of data used in datasets**

| Spend-Ohio dataset | | | | |
|---|---|---|---|---|
| data | entities | distance | relations | records |
| 2022-3 | 39188 | 2km | 2941374 | 1014976 |
| 2022-4 | 39461 | 2km | 2970417 | 1055901 |
| 2022-5 | 39654 | 2km | 3028519 | 1083649 |
| 2022-6 | 39931 | 2km | 3062957 | 1098972 |
| 2023-1 | 41200 | 2km | 3200018 | 1277200 |
| 2023-2 | 41138 | 2km | 3194903 | 1151864 |
| 2023-3 | 42932 | 2km | 3314523 | 1300893 |
| TFNSW dataset | | | | |
| data | entities | distance | relations | records |
| 2015 | 67 | 4km | 496 | 1045200 |
| 2016 | 69 | 4km | 511 | 1212192 |

Specifically, the attributes used in processed Spend-Ohio and TFNSW data are shown in Table 4.

**Table 4: Attributes for constructing SSTKG in datasets**

| Spend-Ohio dataset | |
|---|---|
| attribute | detail explanation |
| placekey | a tuple representing entity location |
| NAICE code | 6-digit code reflecting category |
| temporal records | selling records collected day by day |
| overall records | overall records calculated by past results |
| TFNSW dataset | |
| attribute | detail explanation |
| location | counters' locations |
| suburb | the suburb where counters are located |
| temporal records | traffic volume collected by hour |
| overall records | overall traffic volume aggregated to days/weeks |

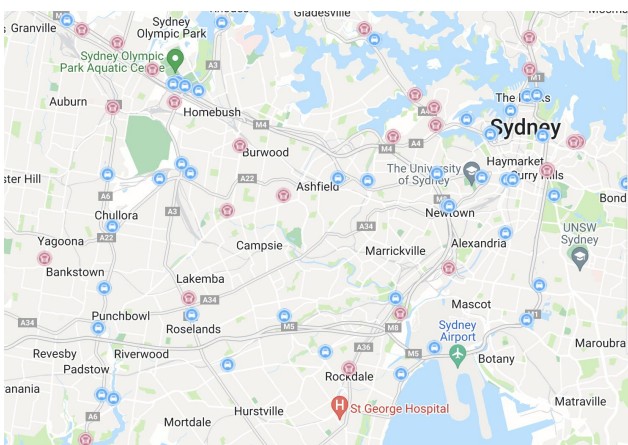

**Figure 3: Part of checkpoints chosen in TFNSW dataset**

### 5.2 Evaluation

The accuracy rate for a prediction ACCn is defined as, if the predicted value is judged as correct, then:

$$r_{predict} \in (r_{real}(1 - n\%), r_{real}(1 + n\%)) \quad (17)$$

When evaluating, apart from accuracy, RMS and RSD are also used for evaluation and defined as:

$$RMS = \sqrt{\frac{\Sigma^n_{i=1}(o_i - p_i)^2}{\Sigma^n_{i=1}(o_i)^2}} \quad (18)$$

$$RSD = \sqrt{\frac{\Sigma^n_{i=1}(o_i - p_i)^2}{N}} \quad (19)$$

Apart from our model (SSTKG), other models are used for comparison: (1) Support Vector Regression Machine(SVR), which is based on support vector machine. (2)Long Short Term Memory (LSTM) network that models the sequence of temporal records. (3) TGCN, the combination of graph convolutional network(GCN) and

**Table 5: Test results for Spend-Ohio datasets**

| method | acc@10 | acc@15 | RMS | RSD |
|---|---|---|---|---|
| Spend-Ohio dataset: 2022.3 - 2022.6 | | | | |
| SVR | 0.5621 | 0.6528 | 0.9872 | 158.9 |
| LSTM | 0.5984 | 0.7025 | 0.9031 | 135.7 |
| GRU | 0.7057 | 0.8544 | 0.607 | 97.3 |
| T-GCN | 0.7489 | 0.8386 | 0.651 | 103.5 |
| ST-GCN | 0.7902 | **0.8945** | 0.463 | 87.9 |
| SSTKG | **0.8016** | 0.8922 | **0.452** | **86.1** |
| Spend-Ohio dataset: 2023.1 - 2023.3 | | | | |
| SVR | 0.6015 | 0.7325 | 0.9751 | 144.3 |
| LSTM | 0.6394 | 0.7672 | 0.8865 | 127.2 |
| GRU | 0.7359 | 0.8897 | 0.528 | 88.3 |
| T-GCN | 0.7826 | 0.8597 | 0.562 | 91.3 |
| ST-GCN | **0.8435** | 0.9291 | 0.399 | 76.8 |
| SSTKG | 0.8374 | **0.9289** | **0.396** | **71.7** |

**Table 6: Test results for TFNSW datasets**

| method | acc@10 | acc@15 | RMS | RSD |
|---|---|---|---|---|
| TFNSW dataset: hourly prediction | | | | |
| SVR | 0.701 | 0.7583 | 0.6737 | 129.8 |
| LSTM | 0.7639 | 0.8072 | 0.5615 | 113.4 |
| GRU | 0.7825 | 0.8404 | 0.475 | 107.8 |
| T-GCN | 0.7973 | 0.8345 | 0.497 | 105.2 |
| ST-GCN | **0.8137** | 0.8641 | 0.429 | 96.9 |
| SSTKG | 0.8095 | **0.8692** | **0.4245** | **95.7** |
| TFNSW dataset: daily prediction | | | | |
| SVR | 0.7914 | 0.8215 | 0.5047 | 90.1 |
| LSTM | 0.8145 | 0.8374 | 0.459 | 87.2 |
| GRU | 0.8609 | 0.9285 | 0.3867 | 63.7 |
| T-GCN | 0.8745 | 0.948 | 0.3641 | 67.5 |
| ST-GCN | 0.8991 | **0.9625** | 0.3583 | **52.8** |
| SSTKG | **0.9051** | 0.9571 | **0.3488** | 54.3 |

gated recurrent unit(GRU), while GCN can learn spatial characteristics of nodes, and GRU learns temporal features of historical temporal records. (4) STGCN [30] which uses 2 TCNs and 1 GCN and could serially capture spatial-temporal dependencies in the data.

### 5.3 Case study

In order to validate the interpretability of the proposed model, a case study was conducted using the Spend-Ohio data in 2023-1. Specific stores served as exemplars. Following the knowledge graph construction and training of the influences and embeddings, the distance thresholds were adjusted to modify the quantity of entities deemed related in the knowledge graph. By repeating the construction process with these variations, differences in outcomes aim to elucidate the model's explainability.

## 6 RESULT

### 6.1 Experiment Results

*6.1.1 Safegraph: Spend-Ohio dataset.* In Spend-Ohio dataset, the first 25 days are used to construct and train the SSTKG for monthly data, while the rest data is used for testing(which is 6 days, 3 days, and 6 days in the three subsets). To help compare and reduce the effect of null values, when calculating the RMS and RSD, the score's selling records is normalized to a range of **(0,20)**. The results are shown in Table 5.

*6.1.2 TFNSW dataset.* In TFNSW data, two separate experiments were done. The first one used the hourly data collected 24/7. 40 weeks' data were used to train, and then a 24-hour prediction in the following days was generated. In the second experiment, hourly records were added to daily ones, then the daily records were used to train. It is similar to the scale in the Spend-Ohio dataset. Similarly, the traffic volume was also normalized to **(0,20)**. The accuracy result (acc10 and acc15) and the RMS and RSD for normalized data are shown in Table 6:

### 6.2 Result analysis

From the results, the prediction of T-GCN, ST-GCN, and SSTKG are much better than SVR and LSTM. This is because SVR and LSTM only focus on temporal record correlations while failing to consider spatial relations. SSTKG, as well as T-GCN and ST-GCN, model both spatial and temporal characteristics to ensure the data effectiveness. SSTKG is better than T-GCN. Compared with ST-GCN, SSTKG performs better on acc15 and RSD on the Spend-Ohio dataset, while being better in acc15, RMS, and RSD for hourly prediction, in acc10 and RMS for daily prediction in the TFNSW dataset.

### 6.3 Interpretability: case study

This section presents the predicted result for a single entity in the Spend-Ohio dataset, in order to demonstrate the interpretability of SSTKG. The selected sample entity possesses attributes outlined in Table 7, with certain values masked to maintain privacy.

**Table 7: Attributes of sample entities**

| attribute | value |
|---|---|
| placekey | 225-222@63j-xxx-xxx |
| NAICS | 722511 |
| Type | Full type service restaurant |

For this entity, distances of nearby entities are shown in Figure 4. There are 36 entities in SSTKG that have influence with this shop. Figure 5 shows the influence values that are calculated and extracted from SSTKG.

From the above results, generally, entities close to the sample are more likely to have larger influence values, whereas those entities far away from samples have nearly no influence.

By integrating the above influences with trained embeddings, the sample's selling is predicted based on Equation (6), (8) and (9)

**Table 8: Comparison of Real values with Predicted and Adjusted Data**

| Day | Real values | Former Predicted data($R_0$) | Remove entity A($R_a$) | Remove entity B($R_b$) | Remove entities C($R_c$) |
|---|---|---|---|---|---|
| Day 1 | 263.95 | 277.47 | 257.44 | 289.69 | 281.82 |
| Day 2 | 495.81 | 530.09 | 517.74 | 539.26 | 528.27 |
| Day 3 | 257.85 | 239.37 | 228.33 | 245.70 | 242.62 |
| Day 4 | 352.82 | 372.83 | 352.14 | 381.54 | 373.38 |
| Day 5 | 196.54 | 188.06 | 172.63 | 191.41 | 190.35 |
| Day 6 | 409.67 | 435.99 | 413.76 | 443.57 | 434.63 |
| Day 7 | 200.7 | 189.11 | 180.15 | 198.97 | 185.34 |

(first calculate embeddings then decode records). However, if some related entities were masked, the predicted result would change, like entities A, B, and group C shown in Figure 5, while entity A has a positive influence, B has a negative influence, and group C has barely non-influence. Table 8 shows the change of prediction after masking entities A, B, and group C.

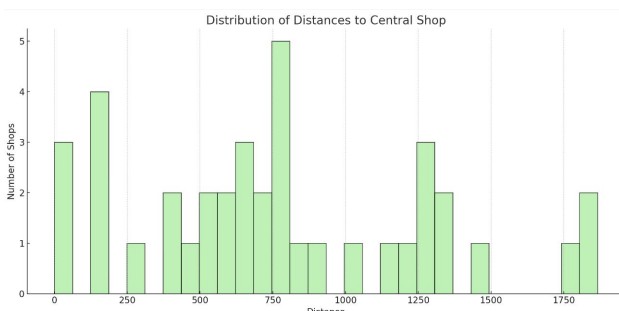

**Figure 4: Related entities' distances with sample shop**

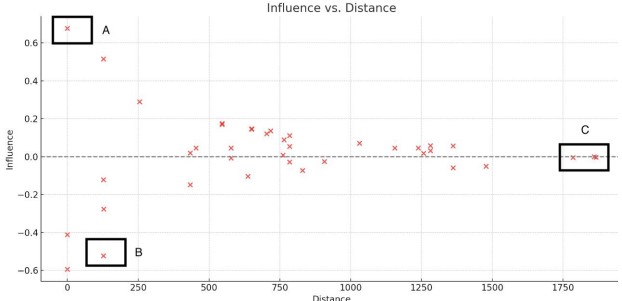

**Figure 5: Related entities' influence to sample shop**

A hypothesis test is set to show the difference between predicted data. While the former predicted data is $R_0$, predicted data after removing A, B and C are $R_a, R_b$ and $R_c$, the null hypothesis are: $H_{0a} : R_0 < R_a$; $H_{0b} : R_0 > R_b$; $H_{0c} : R_0 != R_c$, while alternative hypothesis are $H_{1a} : R_0 > R_a$; $H_{1b} : R_0 < R_b$; $H_{1c} : R_0 = R_c$. Table 9 shows the p-value after t-test under 95% confidence level:

For all three null hypotheses, the p-value of t-test is greater than 0.05, thus are all rejected, drawing the conclusion that, by masking entity A, the predicted value for sample's selling decreased($R_0 > R_a$), while by masking B the predicted value increased ($R_0 > R_b$)

– those who have positive influence on SSTKG would increase prediction, which means "prosperity in one shop leads to prosperity to another", and vice versa. On the other hand, in group C, where entities have small influence values, the prediction value changed a little after masking them (more than 95% confidence to confirm that $R_0 = R_c$).

**Table 9: Result for t-test**

| hypothesis | p-value | result |
|---|---|---|
| $H_{0a} : R_0 < R_a$ | 0.9998975 | reject $H_{0a}$, accept $H_{1a}$ |
| $H_{0b} : R_0 > R_b$ | 0.999873 | reject $H_{0b}$, accept $H_{1b}$ |
| $H_{0c} : R_0 != R_c$ | 0.6717662 | reject $H_{0c}$, accept $H_{1c}$ |

## 7 CONCLUSIONS AND FUTURE WORK

In this paper, a new knowledge graph framework is proposed, i.e., simple spatial-temporal knowledge graph (SSTKG), which leverages 3 kinds of embeddings (static, temporal in and out embeddings) to model entities, as well as using "influence" to model the spatial-temporal relations between entities. A comprehensive evaluation using real-world data has underscored the efficacy of the proposed SSTKG in prediction tasks and highlighted its interpretability. Future endeavors will focus on refining the SSTKG construction algorithm, moving beyond distance thresholds to embrace node similarity without spatial constraints. Moreover, the potential application of SSTKG in recommendation tasks will also be explored.

## 8 ETHICAL USE OF DATA

The Spend-Ohio dataset from SafeGraph was utilized for this study. While it provides granular transaction data, all transactions and associated credit or debit card details have undergone rigorous anonymization to safeguard consumer privacy. Specific details about the merchants (like location and brand) within the Spend-Ohio dataset were masked from the study. All information regarding merchants and consumers was handled with strict confidentiality, ensuring that no privacy boundaries were breached. No credit information of merchants and consumers is involved in this paper.

Additionally, the TFNSW dataset used in the experiment is a publicly available dataset that contains neither personal nor private details. The dataset only incorporates generic traffic flow without identifiable details, without specific identifiable details such as license plate numbers or exact timestamps of certain car passes.

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
