# OpenReview forum: "Exploiting Spatial-Temporal Data in Knowledge Graphs for Enhanced Prediction"
_ACM.org/TheWebConf/2024/Conference — TheWebConf24 Oral_

### Official Review · Reviewer_aGW2 · 2023-11-09

**Novelty:** 4
**Technical Quality:** 2

**Review:**

This paper presents an innovative approach to utilizing spatial-temporal data in knowledge graphs. However, the evaluation is limited by the lack of a comprehensive comparison with the latest state-of-the-art models and validation across diverse datasets.

Pros:\
S1: The paper introduces a novel framework for constructing and exploiting Spatial-Temporal Knowledge Graphs, integrating spatial and temporal data to enhance prediction accuracy and recommendation relevance.\
S2: The model is designed with computational efficiency in mind, which is crucial for fast STKG construction and could be beneficial in real-time data processing applications.\
S3: The paper is well organized.

Cons:\
W1: The primary concern identified is the absence of benchmarking against latest SOTA models. The most advanced baseline referenced in Tables 5 & 6 is ST-GCN, which was published in 2017. Depending on such outdated benchmarks without demonstrating enhancements or benefits compared to newer and more sophisticated models, including ST-GCN itself, undermines the significance of the proposed methodology.\
W2: While the paper aims to address the versatility and efficiency of the framework, it is unclear how well it generalizes across different datasets and its adaptability to various domains.\
W3: Validation on Diverse Datasets: The validation of the framework appears to be limited to two datasets. Additional testing on a broader range of datasets would strengthen the claims of efficiency and accuracy.

**Questions:**

See W1-W3.

**Reviewer Confidence:**

3: The reviewer is confident but not certain that the evaluation is correct

**Scope:**

4: The work is relevant to the Web and to the track, and is of broad interest to the community

---

### Official Review · Reviewer_Pk7f · 2023-11-09

**Novelty:** 4
**Technical Quality:** 5

**Review:**

This paper proposes a knowledge graph framework, simple spatial-temporal knowledge graph (SSTKG), for prediction and recommendation. The framework leverages 3 kinds of embedding (static, temporal in and out embedding) to model entities and exploits “influence” to model the spatial-temporal relations between entities, aiming to leverage the strengths of KGs to enhance the accuracy and relevance of spatial-temporal predictions. Comprehensive evaluations with real-world data have verified the efficacy of the proposed SSTKG in prediction tasks and highlighted its interpretability.

Strengths can be found below.

Strengths:
- S1. It Is a good idea to use knowledge graphs to improve spatial-temporal for prediction and recommendation as KGs can help overcome challenges such as scalability with large datasets and inadequate semantic understanding.

- S2. The proposed methods are evaluated on datasets regarding both traffics and selling records.

**Questions:**

- Q1. In section 3.2, it seems to lack some condition to fully understand algorithm 1. In line 2 of  algorithm 1, maybe anything missing after “where”? If not, it just indicates implement the following filtering in set $E_0$, “where” here may be misleading.

- Q2. Some explanations in the article are unclear and confusing. For example, in the quadruple (e_i,t, t_j, r) , what does t and t_j mean? It was not explained in the manuscript.

- Q3. The article lacks comparison on “complexity of time” with other methods. Considering the proposed SSTKG performs closely similar with STGCN in terms of accuracy and error, it is hard to say which method is better. More explanations on it are required.

- Q4. Experiments are not sufficient.

- Q4-1. The authors only conduct experiments on limited two datasets, although the datasets have covered from traffic to recommendation, they still lack the cross-validation on more real-world scenarios (datasets).

- Q4-2 The compared methods are limited, and lack the SOTA spatiotemporal and recommendation solutions. To name a few,
Spatiotemporal learning: GraphWaveNet[1], ST-SSL[2], AGCRN [3], Recommendation: KGAT[4], KAC[5].

- Q5. The authors still do not outline the specific challenges of embedding spatiotemporal elements into knowledge graph and do not explain why the previous strategies fail to embed the knowledge information. Please do more explanations.

[1] Wu Z, Pan S, Long G, et al. Graph wavenet for deep spatial-temporal graph modeling[J]. arXiv preprint arXiv:1906.00121, 2019.

[2] Ji J, Wang J, Huang C, et al. Spatio-temporal self-supervised learning for traffic flow prediction[C]//Proceedings of the AAAI Conference on Artificial Intelligence. 2023, 37(4): 4356-4364.

[3] Bai L, Yao L, Li C, et al. Adaptive graph convolutional recurrent network for traffic forecasting[J]. Advances in neural information processing systems, 2020, 33: 17804-17815.

[4] Wang X, He X, Cao Y, et al. Kgat: Knowledge graph attention network for recommendation[C]//Proceedings of the 25th ACM SIGKDD international conference on knowledge discovery & data mining. 2019: 950-958.

[5] Wang H, Xu Y, Yang C, et al. Knowledge-Adaptive Contrastive Learning for Recommendation[C]//Proceedings of the Sixteenth ACM International Conference on Web Search and Data Mining. 2023: 535-543.

**Ethics Review Description:**

N/A.

**Reviewer Confidence:**

4: The reviewer is certain that the evaluation is correct and very familiar with the relevant literature

**Scope:**

3: The work is somewhat relevant to the Web and to the track, and is of narrow interest to a sub-community

---

### Official Review · Reviewer_Eia8 · 2023-11-17

**Novelty:** 5
**Technical Quality:** 6

**Review:**

This paper presents a framework that constructs spatial-temporal KGs by integrating spatial and temporal data. It introduces a 3-step embedding method to generate embeddings useful for predicting future temporal sequences and recommending spatial information. Authors argue that most existing methods lean on static data, overlooking the dynamic essence and concealed spatial-temporal characteristics found in real-world situations. The authors conducted an experimental analysis on two benchmark datasets and showed that their framework outperforms state-of-the-art approaches.

Strengths:
* S1: This paper demonstrates a clear examination of the limitations present in current approaches.
* S2: It offers a good overview of related work, showcasing a deep understanding of the existing landscape.
* S3: The utilization of two distinct datasets as benchmarks to evaluate their approach highlights a robust and well-rounded assessment of their method against state-of-the-art approaches.

Weaknesses:
* W1: The authors list several prior approaches in section 2.2.3 that have been proposed to address the same problem as the one being addressed in this paper. However, the authors do not go into detail about how these approaches work. This makes it difficult for the reader to understand the relative merits of the different approaches and to assess the novelty of the proposed approach. To improve clarity, the authors could provide a brief overview of the key ideas behind each approach, as well as a discussion of the strengths and weaknesses. Additionally, a real-world application or example would help to demonstrate the practical value of the proposed approach.

* W2: Considering the substantial scope of the authors' work, having a discussion and limitation section would be valuable. I anticipate there might be several aspects worth exploring in such a section.

**Questions:**

* Q1: I may have overlooked this in the paper, but it wasn't evident to me whether the proposed framework addresses a problem that's exclusively unsolvable by other approaches or if it enhances previous methods. Providing an example of the applications requiring this framework that other methods cannot resolve would be insightful.
* Q2: In Rule 1 within subsection 3.2.2, it's outlined that location and time are not treated as independent entities but are integrated as inherent attributes shared between entities. While I understand the fluidity across both temporal and spatial dimensions for any pair of entities, there might be limitations to this approach, especially concerning location. Can the authors discuss the drawbacks, if any, of this method? It seems to overlook scenarios where individual entities have static relationships with specific locations.

**Reviewer Confidence:**

2: The reviewer is willing to defend the evaluation, but it is likely that the reviewer did not understand parts of the paper

**Scope:**

4: The work is relevant to the Web and to the track, and is of broad interest to the community

---

### Official Review · Reviewer_1jWc · 2023-11-23

**Novelty:** 5
**Technical Quality:** 5

**Review:**

This paper introduces a framework that addresses the limitations of current methods for link prediction and recommendation in knowledge graphs (KGs). By integrating spatial and temporal data, the framework constructs spatial-temporal KGs and employs a new embedding method. This enhances accuracy and relevance in retail sales forecasting and traffic volume prediction applications.

**Questions:**

1. We would recommend you offer your codes and data after the paper is accepted.

2. More experiments should be carried out on different datasets, such as UUKG [1]

[1] UUKG: Unified Urban Knowledge Graph Dataset for Urban Spatiotemporal Prediction

3. Besides TransE, lack of other related literature for KGs, Spatio-temporal graphs, such as [2][3][4][5]

[2] A survey on knowledge graphs: Representation, acquisition, and applications. https://arxiv.org/pdf/2002.00388.pdf

[3] Reasoning over different types of knowledge graphs: Static, temporal, and multi-modal.https://arxiv.org/pdf/2212.05767

[4] RotatE: Knowledge Graph Embedding by Relational Rotation in Complex Space. https://openreview.net/forum?id=HkgEQnRqYQ

[5] Spatial-Temporal Sequential Hypergraph Network for Crime Prediction with Dynamic Multiplex Relation Learning. https://www.ijcai.org/proceedings/2021/0225.pdf

**Reviewer Confidence:**

3: The reviewer is confident but not certain that the evaluation is correct

**Scope:**

3: The work is somewhat relevant to the Web and to the track, and is of narrow interest to a sub-community

---

### Official Review · Reviewer_rcoP · 2023-11-24

**Novelty:** 3
**Technical Quality:** 3

**Review:**

This paper introduces a framework for constructing and exploring spatial-temporal KGs. The authors firstly integrate spatial and temporal data to form KGs. These KGs are further exploited through a new 3-step embedding method. Output embeddings can be used for future temporal sequence prediction and spatial information recommendation.

* Pros:
  * This paper is well organized with clear clarity and thus easy to follow.
  * The application of Spatial-temporal KGs is meaningful in the real world.
* Cons:
  * The contribution of this paper is not clear.
  * More spatial-temporal graph techniques could be the compared baselines to show the effectiveness of the proposed model.
  * The overall writing needs improvement.

**Questions:**

1.  My biggest concern is the confusing contribution of this paper. A “clear” list of contributions should be given at the end of intro. Does the process of constructing spatial-temporal KGs and proposing two new datasets belong to your main contributions? In addition, what contributions does the proposed model have compared to previous works? I’m really confused after reading the paper.
2. The latest baseline is published in 2017. While not an expert in this field, it seems implausible that no advanced spatial-temporal graph techniques have been proposed since 2018 and beyond. This assumption appears to be quite unreasonable.
3. Needs to be revised: A Figure that illustrates the proposed model is needed. Related work Section (especially 2.1/2.2) is too brief. Is the Eq. 7 correct? Two loops in Algorithm 2 are actually the same, which is redundant.

**Reviewer Confidence:**

2: The reviewer is willing to defend the evaluation, but it is likely that the reviewer did not understand parts of the paper

**Scope:**

4: The work is relevant to the Web and to the track, and is of broad interest to the community

---

### Official Review · Reviewer_8fg9 · 2023-11-24

**Novelty:** 5
**Technical Quality:** 4

**Review:**

## Summary
The paper focuses on spatial-temporal Knowledge Graphs (KGs) and highlights the limitations of existing methods. The authors propose a framework for constructing and exploring spatial-temporal KGs that seamlessly integrates spatial and temporal data. The resulting embeddings are utilized for future temporal sequence prediction and spatial information recognition, with the aim of enhancing the utilization of spatial-temporal data in KGs.

## Strengths
+ The paper addresses a meaningful and practical topic.
+ The motivation behind the research is clear.
+ The experiments conducted are extensive.

## Weaknesses
+ The introduction of the authors' work is limited, with only one paragraph dedicated to their method. It would be beneficial to provide a more comprehensive introduction, including a discussion of the background and related work.
+ There appears to be a lack of related literature on spatial-temporal models. It is recommended to conduct a more thorough review of existing research in this field.
+ The paper seems to primarily apply existing methods in a spatial-temporal scenario. It would be helpful for the authors to clearly explain the novelty and contribution of their work, including how it differs from previous approaches or fills a research gap.
+ The baselines used in the experiments are outdated. It would be valuable to compare the proposed method with more recent approaches, such as the Transformer model, as shown in Table 5.
+ The paper lacks experiments on memory cost. It would be beneficial to include this aspect in the evaluation.
+ To enhance reproducibility and contribute to the research community, it is suggested that the authors release the processed KG datasets.

**Questions:**

See weaknesses.

**Reviewer Confidence:**

3: The reviewer is confident but not certain that the evaluation is correct

**Scope:**

3: The work is somewhat relevant to the Web and to the track, and is of narrow interest to a sub-community

---

### Decision · Program_Chairs · 2024-01-22

**Decision:**

Accept (Oral)

**Comment:**

Summary: The paper proposes to incorporate spatio-temporal data in knowledge graphs, purportedly to improve accuracy of prediction of downstream tasks.

 Strengths:
 + well-motivated problem
 + extensive experiments

 Weaknesses:
 - lack of literature coverage of spatio-temporal models
 - can be strengthened with more spatial-temporal graph techniques as baselines

 Reasonable proposition, with potential impact if KG is released.